Metabolic cost of walking with electromechanical ankle exoskeletons under proportional myoelectric control on a treadmill and outdoors

Hybart Rachel rhybart@ufl.edu
Villancio-Wolter K. Siena
Ferris Daniel Perry
J. Crayton Pruitt Department of Biomedical Engineering, University of Florida , Gainesville , FL , United States of America
van den Bogert Antonie
Electronic publication date: 2023 Jul 27
Publication date: 2023
Volume: 11
Electronic Location ID: e15775
Received 2023 Apr 13; Accepted 2023 Jun 29
Copyright: ©2023 Hybart et al.
Copyright year: 2023
Copyright holder: Hybart et al.
License: This is an open access article distributed under the terms of the Creative Commons Attribution License, which permits unrestricted use, distribution, reproduction and adaptation in any medium and for any purpose provided that it is properly attributed. For attribution, the original author(s), title, publication source (PeerJ) and either DOI or URL of the article must be cited.
License URL: https://creativecommons.org/licenses/by/4.0/

Keywords: Exoskeleton, Metabolics, Myoelectric, Lower-limb

Funding: National Institute of Health R01 NS104772 This work was supported by the National Institute of Health under grant R01 NS104772. The funders had no role in study design, data collection and analysis, decision to publish, or preparation of the manuscript.

==============================
Lower limb robotic exoskeletons are often studied in the context of steady state treadmill walking in a laboratory environment. However, the end goal for exoskeletons is to be used in real world, complex environments. To reach the point that exoskeletons are openly adopted into our everyday lives, we need to understand how the human and robot interact outside of a laboratory. Metabolic cost is often viewed as a gold standard metric for measuring exoskeleton performance but is rarely used to evaluate performance at non steady state walking outside of a laboratory. In this study, we tested the effects of robotic ankle exoskeletons under proportional myoelectric control on the cost of transport of walking both inside on a treadmill and outside overground. We hypothesized that walking with the exoskeletons would lead to a lower cost of transport compared to walking without them both on a treadmill and outside. We saw no significant increases or decreases in cost of transport or exoskeleton mechanics when walking with the exoskeletons compared to walking without them both on a treadmill and outside. We saw a strong negative correlation between walking speed and cost of transport when walking with and without the exoskeletons. In the future, research should consider how performing more difficult tasks, such as incline and loaded walking, affects the cost of transport while walking with and without robotic ankle exoskeletons. The value of this study to the literature is that it emphasizes the importance of both hardware dynamics and controller design towards reducing metabolic cost of transport with robotic ankle exoskeletons. When comparing our results to other studies using the same hardware with different controllers or very similar controllers with different hardware, there are a wide range of outcomes as to metabolic benefit.

Introduction

Lower limb robotic exoskeletons are used in a variety of real-world applications. Therapeutic training, physical assistance, and human augmentation are the usual goals for lower limb robotic exoskeleton design (Young & Ferris, 2017). In rehabilitation, exoskeletons can allow neurologically impaired patients to practice movements that they could not complete without assistance and can also serve as assistive technology to replace lost physical capabilities (Milia et al., 2018). For able-bodied humans, robotic lower limb exoskeletons can assist with physically demanding tasks like lifting heavy objects safely, walking with heavy loads and extending endurance for walking long distances (MacLean & Ferris, 2019; Yandell, Tacca & Zelik, 2019; Maurice et al., 2020; Tang et al., 2022). Current pitfalls of robotic exoskeletons include that they are often too heavy, bulky, or require extensive adaptation by the user (Young & Ferris, 2017; Fritz, Patzer & Galen, 2019; van Dijsseldonk et al., 2020; Siviy et al., 2022).

A common method of quantifying the effectiveness of robotic lower limb exoskeletons is by evaluating the changes in metabolic cost (Sawicki et al., 2020). Exoskeletons ideally should be able to decrease the metabolic cost when wearing the powered device compared to not wearing the device or wearing the device in an unpowered, transparent mode (Au, Weber & Herr, 2009; Grabowski & Herr, 2009; Mooney, Rouse & Herr, 2014; Collins, Bruce Wiggin & Sawicki, 2015; Galle et al., 2017; Seo et al., 2018; Panizzolo et al., 2019). Previously published studies on powered ankle exoskeletons have shown a decrease in metabolic cost ranging from 5.6–21.4% during powered walking compared to unpowered walking or not wearing the device (Mooney, Rouse & Herr, 2014; Galle et al., 2017; Sawicki et al., 2020). There are currently very few lower-limb exoskeleton studies that have measured metabolic cost outside of the laboratory setting. MacLean & Ferris (2019) evaluated a robotic knee exoskeleton and found that although it provided a 4.2% decrease in metabolic cost during indoor incline walking on a treadmill with a backpack, it increased metabolic cost of walking on a hilly terrain course outdoors (MacLean & Ferris, 2019). A recent study was able to achieve a 17% decrease in the metabolic cost of walking using an untethered ankle exoskeleton by optimizing the control of the ankle exoskeleton for each individual (Slade et al., 2022). The inclusion of outdoor walking in these studies is an important step towards understanding how robotic exoskeletons affect metabolic cost in real-world environments. While indoor treadmill walking provides steady-state conditions, it cannot accurately reflect all aspects of a real-world environment. Outdoor trials reflect the presence of obstacle avoidance, variable walking speed, changes in direction, and uneven terrain (Parker et al., 2021; Schmitt et al., 2021; Semaan et al., 2022; Soliman et al., 2022).

A very important aspect determining the success of robotic lower limb exoskeletons is the control method. Many studies have used state-based controllers, which sense kinematic and kinetic information about the gait cycle and the environment in order to determine the control pattern (Masengo et al., 2020; Sawicki et al., 2020; Hybart & Ferris, 2022; Lora-millan, Moreno & Rocon, 2022; Siviy et al., 2022). This approach can be very successful for flat, level terrain when walking at a constant speed. In those settings, the next few steps are very much like the last few steps. Another alternative to kinematic and kinetic state control is to use neural signals from the human by incorporating electromyography (EMG) or electroencephalography signals (Ferris & Lewis, 2013; He et al., 2018; Tariq, Trivailo & Simic, 2018). A recent study using an adaptive controller based on EMG and joint angle found a decrease in the metabolic cost of walking by 22% when compared to the Unpowered, transparent condition (Jackson & Collins, 2019).

The purpose of this study was to determine the effect of proportional myoelectric-controlled, robotic ankle exoskeletons on metabolic cost of walking indoors on a treadmill and outdoors overground. We altered the controller on commercially available robotic ankle exoskeletons (Dephy Exoboots) to implement a proportional myoelectric controller similar to what has been used in many treadmill studies (Gordon & Ferris, 2007; Sawicki & Ferris, 2008; Sawicki & Ferris, 2009a). The exoskeletons aided plantarflexion by supplementing torque during the push off phase of stance based on the user’s soleus EMG amplitude (Hybart & Ferris, 2022). This proportional myoelectric control allows the person wearing the device to have intentional control over the exoskeleton’s activation and deactivation. Past studies using pneumatic ankle exoskeletons under proportional myoelectric control for walking on a treadmill found large decreases in metabolic cost (Sawicki & Ferris, 2008; Sawicki & Ferris, 2009a; Sawicki & Ferris, 2009b). We hypothesized that the metabolic cost of walking while wearing the powered exoskeleton would be lower compared to not wearing the exoskeletons and compared to unpowered walking in both laboratory and real-world environments. By using electromechanical ankle exoskeletons, the participants were untethered and could freely walk on a university campus setting outdoors in this study. We were able to compare changes in metabolic cost for indoor treadmill walking and outdoor overground walking.

Materials & Methods

Participants

We recruited 12 participants (five female, seven male) with the following characteristics, mean (s.d.): age 22.6 (8.66) yrs.; height: 1.73 (0.075) m; mass: 69.3 (10.2) kg The standard deviation for the age is large because the minimum and maximum ages were 18 and 50 years, respectively. Without the 50-year-old, the mean and standard deviation in age were 20.1 (2.1). Other data from the 50-year-old participant was not an outlier and was therefore kept in the analysis. Participants had no previous neurological or musculoskeletal conditions. Each participant was right-handed and had no previous experience walking with a robotic ankle exoskeleton prior to the first training session for this study. The University of Florida Institutional Review Board (IRB201801218) approved this study. Each participant read and signed an informed consent form approved by the University of Florida Institutional Review Board (IRB201801218). We expected 12 participants would provide sufficient statistical power based on our previous research (Sawicki & Ferris, 2008; Sawicki & Ferris, 2009a; Koller, Remy & Ferris, 2018; MacLean & Ferris, 2019). Preliminary statistical power analysis on pilot data for three participants indicated that 12 participants would provide 0.8 power for an alpha equal to 0.05.

Equipment

We used the commercially available Dephy exoskeletons (EB60) with an open-source software option, allowing us to implement our own high-level control (Dephy, Inc. Maynard, MA). A proportional myoelectric controller with an input of the user’s soleus muscle provided the input current to the motor (Hybart & Ferris, 2022). The soleus electromyography (EMG) data were high-pass filtered (2nd order Butterworth, cutoff frequency 50 Hz), full wave rectified, and low-pass filtered (2nd order Butterworth, cutoff frequency 8 Hz). We then multiplied the resulting signal by a participant-specific gain determined while walking with the exoskeletons unpowered to get the motor current (Fig. 1). During the Unpowered condition, the soleus EMG were processed through the same pipeline as during the Powered condition in real time. The output of this pipeline was a target current. This target current was only sent to the exoskeleton as a controlling current during the Powered condition, but the researchers were able to visualize this output during all conditions. During the Unpowered condition the participant-specific gains were determined to ensure that the target current reached the maximum value of 7.6 A q-axis (phase) current, which is equivalent to 20 A of line-to-line current as described by Dephy, Inc. This target current was then sent as a controlling current during the Powered condition. The controlling current was transformed via a variable transmission ratio into an ankle torque that pulled the toe down. The exoskeletons only assisted in plantarflexion and the control signal dropped to zero during swing when the foot switches did not register any weight on the foot. The only resistance to dorsiflexion during swing was the passive unwinding of the chain and the weight of the boots. The total system added 4.5 kg to the participants.

Figure 1 Dephy Exoboots worn by participant and an example EMG and control signal.

Dephy Exoboots with Coapt EMG bipolar sensors visible on the tibialis anterior muscles. Example of raw and filtered soleus EMG and the controlling current that applies a plantarflexor torque at push off.

We measured EMG on the soleus, gastrocnemii, and tibialis anterior using bipolar skin electrodes (Coapt, Inc. Chicago, IL, USA). We only used the soleus EMG in the control of the exoskeletons. We used force sensitive resistors to determine strides while outside and inertial measurement units (IMU) to find ankle angle (APDM, Inc. Portland, OR, USA). We used a force instrumented treadmill to determine indoor gait events (Bertec, Co. Columbus, OH, USA). Participants wore a facemask to take respiratory measurements (Cosmed K5, Rome, Italy). The metabolic system added an additional 0.9 kg to the participants. We asked participants not to eat for at least 3 h prior to the data collection to reduce variability in metabolic cost measurements.

Collection

Prior to this collection, participants completed three separate training days of 30 min each walking with the robotic ankle exoskeletons. Participants stood for 6 min to determine their standing baseline metabolic cost. We subtracted this baseline measurement from all metabolic cost measurements to determine the net metabolic cost for each condition. Participants completed five total walking trials (three indoor and two outdoor) (Fig. 2A). The trials were semi-randomized with all indoor trials completed at one time and both outdoor trials completed back-to-back to reduce time spent travelling between testing areas. For the indoor trials, participants walked for 8 min at a self-selected speed, determined on the first day of training, on a force instrumented treadmill. The indoor conditions were Boots Only, Unpowered, and Powered. In the Boots Only condition participants wore the boots used with the exoskeletons but did not wear the exoskeletons. In the Unpowered condition participants wore the exoskeletons in a transparent mode. In the Powered condition the exoskeletons were on and assisting in plantarflexion. For the outdoor conditions, a human pacer walking next to the participants kept participants at a self-selected speed chosen on the treadmill via verbal feedback. The metabolic system stored speed data and heartrate data from a chest strap monitor that was time synced to the metabolic data. For the outdoor conditions, we omitted the Unpowered condition to reduce the length of the total data collection. The outdoor trials consisted of the participants walking a 0.45 km loop three times (Fig. 2B). Participants briefly paused at the end of each lap to mark the laps in the metabolics and exoskeleton systems. After each condition, participants rested to return to baseline metabolic cost before beginning the next trial.

Figure 2 The collection protocol and map of the outdoor course that partcipants walked.

(A) Diagram of all five conditions. We randomized conditions within the indoor and outdoor conditions, as well as between conditions. All indoor and all outdoor conditions occurred consecutively to reduce time spent travelling between indoor and outdoor test sites. (B) Map of the outdoor route located on the University of Florida campus. Each lap was 0.45 km for a total of 1.35 km per condition. The green dot indicates the start and stopping point.

Analysis

We calculated the metabolic cost using the Brockway equation (Brockway, 1987). We filtered the outdoor metabolic cost data with a fourth-order, low- pass, Butterworth filter (cutoff frequency of 0.1 Hz). We calculated the net metabolic cost for each condition by subtracting the standing baseline metabolic cost. For indoor conditions, we averaged the last 3 min of steady-state metabolic data to determine the metabolic cost for that condition. We present the data as cost of transport to account for differences in speed between participants. We show the outdoor data as the timeseries because participants did not reach a steady state during those trials.

Statistical analysis

For the indoor data we performed a one-way ANOVA to look at differences in cost of transport between our three conditions. For the outdoor conditions, we analyzed the filtered metabolic data as a time series and performed a t-test using statistical parametric mapping (SPM) (Pataky, 2010; Donnelly et al., 2017).

Results

Contrary to our hypothesis, there were no significant differences in the cost of transport across conditions (ANOVA, p = 0.20) during indoor treadmill walking. The cost of transport for the Boots Only condition was 2.8 ± 0.60 J kg−1m−1, for the Unpowered condition it was 3.2 ±  0.50 J kg−1m−1, and for the Powered condition it was 3.1 ± 0.60 J kg−1m−1 (Fig. 3). There was a 16% larger average cost of transport in the Unpowered condition compared to the Boots Only condition, an 11% larger average cost of transport in the Powered condition compared to the Boots Only condition and a 5% lower average cost of transport in the Powered condition compared to the Unpowered condition. The average heart rate for the Boots only condition was 95 ± 15 bpm, for the Unpowered condition it was 97 ± 18 bpm, and for the Powered condition it was 96 ± 17 bpm (ANOVA, p = 0.94).

Figure 3 Cost of transport while walking inside on a treadmill with the exoskeletons powered, unpowered, and without the exoskeletons.

Violin plots showing the net cost of transport (J kg −1 m −1) for the final 3 min of indoor treadmill walking in each condition. The average cost of transport was highest in the unpowered condition (orange) and lowest in the boots only condition (red). The Powered condition (blue) had lower cost of transport than the Unpowered condition, but higher than the Boots Only condition. None of the differences were significant. Within each plot the box indicates the median and first and third quartiles. The shaded area shows the distribution of the data across participants.

Between the outdoor Boots Only and Powered conditions, there were no significant differences in cost of transport (Fig. 4A) or heart rate. There were no significant differences within each condition from lap 1 to lap 3. The average cost of transport across all laps with the exoskeletons was 3.15 W kg−1, while the average cost of transport across all laps without the exoskeletons was 2.79 W kg−1. This is a 12.9% increase in the cost of transport when wearing the exoskeletons compared to not wearing the exoskeletons. The maximum cost of transport when wearing the exoskeletons was 4.49 W kg−1 and the minimum was 2.32 W kg−1. While not wearing the exoskeletons the maximum cost of transport was 3.62 W kg−1 and the minimum was 2.11 W kg−1. Peaks in cost of transport of each lap were at 7–10% of the lap. These correspond to timepoints shortly after the speed of participants dropped to the lowest point (Fig. 4B), and participants just completed the longest uphill portion of the course (Fig. 4C). The minimum cost of transport for each lap was around 90%, which occurred after the longest portion of the course that did not involve a turn. There was a significant negative correlation between participant walking speed and metabolic cost of transport in the boots only condition (p = 0.0013, r2 = 0.8125) and a nonsignificant negative correlation in the Powered condition (p = 0.0618, r2 = 0.5536).

Figure 4 Net cost of transport, average speed, and elevation changes for the outdoor conditions.

(A) Net cost of transport (J kg −1 m −1) for Boots Only (red) and Powered (blue) conditions over the entire 3 laps of the outdoor course. No significant differences were seen over the three laps. Shaded areas represent one standard deviation. (B) Average speed (m/s) over the duration of the course for the Boots Only (red) and Powered (blue) conditions. Shaded areas represent one standard deviation. (C) Change in elevation over the duration of the course. Vertical black lines indicate the starts of lap 2 and 3. Vertical dotted lines indicate the maximum cost of transport within each lap and vertical dashed lines indicate the minimum cost of transport within each lap for the Boots Only (red) and Powered conditions (blue).

During the outdoor Powered condition, there were nonsignificant reductions in peak exoskeleton mechanical power of 12% (p = 0.99), peak exoskeleton torque of 4.2% (p = 0.86), and positive work of 6.0% (p = 0.94) from lap 1 to lap 3. The ankle angle remained consistent between laps (Fig. 5). Normalized soleus muscle activity had no significant changes between lap 1 (root mean square = 0.91), lap 2 (root mean square = 0.90), and lap 3 (root mean square = 0.91) (p = 0.70).

Figure 5 Average exoskeleton mechanical power, work, and torque and participant ankle angle across the three laps of the outdoor trials.

Average Ankle angle (deg), exoskeleton mechanical power (W kg −1), exoskeleton torque (Nm kg −1), and exoskeleton work (J kg −1) for lap 1 (red), lap 2 (orange) and lap 3 (blue) of the outdoor powered condition. Work is separated into positive, negative, and net values. There was a nonsignificant drop in exoskeleton power, torque, and positive work during the final lap of the outdoor condition. The x axis is 0–100% of the gait cycle, with 0% and 100% being heel strikes. Shaded areas represent one standard deviation.

The exoskeleton had a slightly larger but nonsignificant difference in peak mechanical power of 16% (p = 0.48), peak torque of 7.0% (p = 0.057), and positive work of 25% (p = 0.22) during the treadmill condition compared to the outdoor condition (Fig. 6). Normalized soleus muscle activity was significantly lower in the outdoor Powered condition (root mean square = 0.91) than the indoor Powered condition (root mean square = 1.07) (p = 0.017).

Figure 6 Average exoskeleton mechanical power, work, and torque and participant ankle angle when walking on a treadmill compared to outside overground.

Average ankle angle (deg), exoskeleton mechanical power (W kg −1), exoskeleton torque (Nm kg −1), and exoskeleton work (J kg −1) for the outdoor (yellow) and indoor (green) powered conditions. Work is separated into positive, negative, and net values. There was a nonsignificant drop in exoskeleton power and torque during the outdoor condition. The x axis is 0–100% of the gait cycle, with 0% and 100% being heel strikes. Shaded areas represent one standard deviation.

Discussion

Contrary to our hypothesis, we did not find a reduction in the cost of transport when walking with the exoskeleton powered either inside or outside. During the indoor trials there was a nonsignificant increase in cost of transport from the boots only condition compared to both the Unpowered and Powered exoskeleton conditions. There was a nonsignificant decrease from the Unpowered condition compared to the Powered exoskeleton condition, which suggests that the exoskeleton was able to counteract some of the increased metabolic cost due to the added distal mass of the exoskeletons (Browning et al., 2007). In the outdoor conditions, there were no significant differences in cost of transport at any point in the three laps when wearing the exoskeletons compared to not wearing them. There was a cyclical increase and decrease in the metabolic cost during each of the three laps in both conditions at approximately 7–10% of each lap (Fig. 4A). This occurred shortly after participants were at their lowest walking speed, an average of 0.97 m/s. We found a negative correlation between walking speed and metabolic cost of transport. This is supported by previous research that shows when walking slower or faster than their preferred walking speed on level ground, individuals have an increase in cost of transport (Fig. 4B) (Ralston, 1958; Margaria, 1976; Zarrugh & Radcliffe, 1978). When wearing the exoskeleton participants had a less negative correlation between their walking speed and metabolic cost. However, this is most likely due to the already increased metabolic cost due to using the device. The reduced speed occurred around the longer, more gradual of the two curves on the course. Before this curve in the course, at about 90% of each lap, the outdoor cost of transport was at its minimum for each lap. This minimum occurred after the longest portion of the course without any changes in direction, and the longest slightly downhill portion of the course (4 C). The speed was over 1.0 m/s during the portion of the laps that saw a decrease in metabolic cost. There was a strong negative correlation between the walking speed and cost of transport in the boots only condition. There was also a relatively strong negative correlation between walking speed and cost of transport in the Powered condition, but it did not reach statistical significance.

There were no significant differences in exoskeleton power, torque, or work between the indoor and outdoor conditions. A human pacer walked with the participants to help maintain a similar speed overground compared to their speed on the treadmill. The use of the pacer may have produced more consistent strides in the overground walking condition than is typically seen during unpaced overground walking (Schmitt et al., 2021). When walking outside overground and inside on a treadmill, the exoskeletons provided an average mechanical power of 0.45 W kg−1 and 0.54 W kg−1 respectively (Fig. 6).

Other studies using proportional myoelectric control reported larger reductions in metabolic cost and greater exoskeleton mechanical power outputs. In past studies from our laboratory, pneumatic ankle exoskeletons have generated an average peak exoskeleton mechanical power of 1.47 W kg−1 when walking on a treadmill at a similar speed (Gordon & Ferris, 2007; Sawicki & Ferris, 2008; Koller et al., 2015; Koller, Remy & Ferris, 2018). In these studies, they recorded an average reduction in metabolic cost of 14.4% at the end of the Powered condition compared to the Unpowered condition (Sawicki & Ferris, 2008; Koller et al., 2015; Koller, Remy & Ferris, 2018). Comparatively, we saw a reduction of 5% between the Unpowered condition and Powered condition in the study presented in this article. In the other studies using proportional myoelectric control, participants walked at an average speed of 1.21 m s−1, which is slightly higher than the average speed of 1.15 m s−1in the study presented in this article. The lower power output of the exoskeletons in our study compared to previous studies using a similar controller is one likely reason we did not see reductions in the cost of transport. The electromechanical Exoboots used in our study had different mechanical capabilities compared to the pneumatic ankle exoskeletons used in the previous studies (Gordon, Sawicki & Ferris, 2006; Lee, Pan & Rouse, 2019).

In comparison to previous research studies measuring metabolic cost of transport during walking with Dephy ExoBoots, we found a smaller metabolic benefit than shown in those studies. Two other studies used the Dephy ExoBoots under different state-based controllers and recorded metabolic cost of transport (Medrano, Thomas & Rouse, 2022; Shepherd et al., 2022). Medrano, Thomas & Rouse (2022) used a current based controller to apply plantarflexor torques with varying onset times, magnitudes, and durations to determine perceivable metabolic cost changes. Participants in this study walked at 1.25 m s−1. The average metabolic cost reduction they achieved was 12.6% compared to the average metabolic cost of walking without an exoskeleton. They did not measure exoskeleton mechanical power or work but had a peak torque between 12 and 20 Nm. Shepherd et al. (2022) used a convolutional neural network gait phase estimator to provide a peak torque at 84% of stance phase at variable speeds. Participants in this study walked at the sinusoidally varying speeds of 1.1 to 1.6 m s−1 with a 30 s period. The average reduction in metabolic cost they achieved was 5.2% during variable speed walking compared to not wearing the exoskeleton. They did not present exoskeleton mechanical power or work but had a peak torque of 30 Nm. Comparatively, we saw an average peak torque of 17 Nm on the treadmill. One cause of our low metabolic cost benefit is likely that our peak torque is on the lower end of what other studies have achieved using the same hardware. Another cause might be that our controller applied torques earlier in the gait cycle and over a longer duration than the other two papers. Medrano, Thomas & Rouse (2022) had a control duration of 10–60% of the gait cycle with an onset at 25–50% of the stride. Shepherd et al. (2022) applied torques for 38% of the stride with an onset at about 30%. Our applied torque onset was earlier and stayed on longer, with our onset being at 10% of the gait cycle, and the duration being 55–60% of the stride on average. An earlier onset is typical in exoskeletons under proportional myoelectric control because it is directly responding to the physiological signals. The triceps surae muscles are active throughout the stance phase (Sawicki & Ferris, 2008; Koller et al., 2015). The use of proportional myoelectric control in combination with a hardware system that has a nonlinear relationship between muscle input and torque output may lead to a non-ideal relationship between onset timing, duration, and magnitude of actuation. Future implementations of proportional myoelectric control with electromechanical motor could employ state-based gains to correct this nonlinear issue.

Comparison of data from many robotic ankle exoskeleton studies indicate there is not a universal relationship between soleus muscle activity, metabolic cost, and exoskeleton mechanical power when walking with assistive ankle exoskeletons. Various hardware and controller designs provide an assortment of metabolic reductions during gait. Figure 7 shows reduction in metabolic cost and reduction in soleus muscle activity across six different studies. Each study shows the changes in metabolic cost and muscle activity relative to the exoskeleton mechanical power provided in the study. The studies include both exoskeletons under proportional myoelectric control (red) (Sawicki & Ferris, 2008; Koller et al., 2015; Koller, Remy & Ferris, 2018), and exoskeletons under state-based controllers (blue) (Galle et al., 2017; Koller, Remy & Ferris, 2018; Jackson & Collins, 2021). Each of the studies included walking on a treadmill with the exoskeletons powered compared to unpowered. We did not include the original Dephy ankle exoskeleton studies by Mooney and colleagues in Fig. 7 as they did not include mechanical power provided by the exoskeletons during unloaded walking (Mooney, Rouse & Herr, 2014; Mooney & Herr, 2016). For proportional myoelectric control studies, there was a strong relationship between exoskeleton mechanical power and reductions in metabolic cost (red solid line, R2 = 0.97). However, there was not a strong relationship between exoskeleton mechanical power and reductions in soleus muscle activity (red dashed line, R2 = 0.0001). For state-based control studies, there were very weak correlations between exoskeleton mechanical power and reductions in metabolic cost reduction (blue solid line, R2 = 0.03), and between exoskeleton mechanical power and reductions in soleus muscle activity (blue dashed line, R2 = 0.06). The variability across studies suggests that factors other than ankle exoskeleton mechanical power and type of controller are important to determining metabolic cost savings when walking with robotic ankle exoskeletons. The added mass, comfort of fit, and mechanical power transmission efficacy could all play a role in metabolic power reductions when walking with ankle exoskeletons. One previous study examining biomechanical changes to gait with robotic ankle exoskeletons found that subjected learned how to reduce muscle mechanical work at the hip joint during walking to rely more on ankle mechanical power with extended practice (Koller et al., 2015). It may be necessary to customize the controller for each individual’s biomechanical walking pattern to achieve optimal metabolic reduction with robotic exoskeleton assistance (Poggensee & Collins, 2021; Slade et al., 2022).

Figure 7 Comparison of the changes in metabolic cost and soleus RMS with respect to exoskeleton mechanical power between this study and previous literature.

The exoskeleton mechanical power (W/kg) versus the percent reduction in metabolic cost of walking with robotic ankle exoskeletons compared to walking with the exoskeletons unpowered in six different studies (left) and the percent soleus RMS reduction (right). Metabolic cost reduction (solid circles) and soleus RMS reduction (open circles) for four studies, including the one presented in this article, use proportional myoelectric control (red) and three studies use state-based exoskeleton control methods (solid blue). There was a strong linear relationship between power output and metabolic cost when using proportional myoelectric control (red solid line, R2 = 0.97), but not when using other types of control (blue solid line, R2 = 0.03). There was not a strong correlation between the power output and soleus RMS when using proportional myoelectric control (R2 = 0.0001) or other types of controllers (R2 = 0.06).

Based on previous research using robotic ankle exoskeletons, 90 min of training over three training sessions prior to the data presented in this article was sufficient for participants to adapt fully to the assistance (Sawicki & Ferris, 2009a; Malcolm et al., 2013; Slade et al., 2022). There were no significant changes in the exoskeleton power, torque, or work from lap 1 to lap 3 in our study. There also were no significant changes in soleus muscle activation from lap 1 to lap 3. It is possible that many days or weeks of practice with the ankle exoskeletons could have reduced metabolic cost further (Huang, Kram & Ahmed, 2012; Poggensee & Collins, 2021), but our results suggest they were fairly steady state at the time of outdoor testing.

There were several limitations to this study. One limitation is that we did not collect joint angles other than the ankle, nor did we record biological joint moment, power, or work from the knee and hip. Work at the hip is more energetically costly than work at the ankle (Kuo, 2001; Lewis & Ferris, 2008). Therefore, quantifying joint angle, power, moment, and work changes at the hip would provide more insight into the lack of change in the cost of transportation. A second limitation is that we did not include the Unpowered condition outside. We chose not to include this condition to keep the duration of the collection to a reasonable length, especially when asking participants to fast beforehand. Our indoor Unpowered condition had the highest cost of transport of the three conditions. If we had collected an Unpowered outdoor condition, we may have been able to better discern what portion of the metabolic cost while wearing the exoskeletons powered was due to the added mass. Another limitation is that we tried to control the speed of walking overground. Walking at nonpreferred walking speeds increases the cost of transport (Browning & Kram, 2005). Because participants chose their walking speed on the treadmill and we used that same speed overground, it may not have been reflective of their preferred outdoor overground walking (Malatesta, Canepa & Menendez, 2017). We chose to try to speed match the indoor and overground conditions to facilitate comparisons between conditions, but future studies should allow more variability in walking speeds to better represent real world cost of transport. It is important to note that even during steady-state walking on level ground, other studies show fluctuations in cost of transport over time (Zukowski et al., 2017; Wong, Selinger & Donelan, 2019). A final limitation is that we did not take any qualitative measures of performance. Despite metabolic cost being a gold standard for exoskeleton performance, it does not provide the full picture on how the human perceives the device. It is important to take into account how the human embodies assistive devices (Hybart & Ferris, 2022). Using a qualitative or quantitative measure of embodiment and user perception of the device may provide important information on why exoskeletons are not used more often.

The ankle exoskeletons and proportional myoelectric controller used in the study may have fared much better if participants had to traverse inclines or carry heavy backpack loads. Walking up an incline or carrying a load substantially increases the mechanical work and metabolic cost of walking. The ankle exoskeletons may have provided a higher relative benefit in those scenarios. Other lower limb exoskeleton studies have shown decreases in metabolic cost with a load or at an incline (Sawicki & Ferris, 2009a; Mooney, Rouse & Herr, 2014; MacLean & Ferris, 2019). Future studies should include incline and decline walking, walking with a load, and stair ascent and descent, as well as increasing the power output of the exoskeletons to increase the metabolic benefit. Testing robotic exoskeletons over a wide array of conditions and tasks provides a better evaluation of their efficacy in real world scenarios.

Conclusions

We tested the hypothesis that using commercially available robotic ankle exoskeletons under proportional myoelectric control would lead to a reduction in the cost of transport of walking both inside on a treadmill and outside overground when compared to not using the exoskeletons. We found no significant changes in the cost of transport on the treadmill or outside. There were no significant differences in exoskeleton torque or power from the first lap and last laps in the outdoor walking. The lack of change in cost of transport and exoskeleton mechanics across conditions may have been because of too low mechanical power output from the exoskeletons, but comparison with the literature suggests exoskeleton mechanical power is not the driving factor in metabolic cost reductions during walking. One important finding of the study was that results in metabolic cost and gait mechanics from treadmill walking were very similar to overground walking when elevation changes were limited and speed was controlled. Future studies evaluating robotic ankle exoskeletons should consider greater variation in outdoor terrain such as inclines and stairs as they are likely to require increased exoskeleton power and metabolic demand.

We would like to thank Caleb Leimer for his assistance with data collections.

Additional Information and Declarations

Competing Interests

Author Contributions

Human Ethics

Data Availability

The authors declare there are no competing interests.

Rachel Hybart conceived and designed the experiments, performed the experiments, analyzed the data, prepared figures and/or tables, authored or reviewed drafts of the article, and approved the final draft.

K. Siena Villancio-Wolter performed the experiments, analyzed the data, authored or reviewed drafts of the article, and approved the final draft.

Daniel Perry Ferris conceived and designed the experiments, authored or reviewed drafts of the article, and approved the final draft.

The following information was supplied relating to ethical approvals (i.e., approving body and any reference numbers):

The University of Florida granted approval to carry out this study at this facility (IRB 201801218)

The following information was supplied regarding data availability:

The data are available at Figshare:

Hybart, Rachel (2023). Electromyography and ExoBoot Proportional Myoelectric Control Data. figshare. Dataset. https://doi.org/10.6084/m9.figshare.22572868.v3.

Hybart, Rachel (2023). Metabolics Data: Proportional Myoelectric Control of Dephy ExoBoots. figshare. Dataset. https://doi.org/10.6084/m9.figshare.22572841.v4.

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
