# Peer review of "Metabolic cost of walking with electromechanical ankle exoskeletons under proportional myoelectric control on a treadmill and outdoors"

_PeerJ, doi:10.7717/peerj.15775_

## Round 0.1 · original submission · Minor Revisions

The reviewers commented that the manuscript is well written and the study is interesting and sound. They also provided some constructive comments, which should help revise the manuscript.

Reviewer 1 ·

Basic reporting

• Line 66-68: “In rehabilitation, exoskeletons can allow neurologically impaired patients to practice movements that they could not complete with assistance…” Comment: Should this be “…could not complete without assistance…”? I don’t understand the syntax otherwise.
• Figure 4 needs a horizontal axis label.

Experimental design

• Line 76: The term “transparent” is used here for the first time. Broadly, how transparent is ‘transparent’? Specifically, how much energy is lost just through the (passive) actuation of the device while walking with this transparent mode? I understand that the energy loss should be close as close to zero as possible, but it can’t be zero. It would be good to include some sort of confirmation that this energy loss is negligible when compared to the overall metabolic cost of walking.
• Materials & Methods, Equipment: Here you discuss how the EMG signal was processed, and reference Figure 1, which includes an example of the “Control Signal.” Is this final control signal just multiplied by the subject-specific gain? Or was other manipulation (eg. muscle force dynamics) applied as well?
• Do you have an elevation profile for the outdoor course? High cost of transport during their slowest speed makes me think that the subjects are going uphill at that point, but you note in the Discussion that this point occurred during a turn. It would be helpful to plot Figures 4 and 5 on the same plot, along with an elevation profile, and possibly some sort of radius of curvature of the turns on the course so that peaks/valleys can be easily lined up. Possibly include vertical dashed lines to indicate local maxima/minima.
• Was each outdoor lap completed consecutively, or did subjects stop and restart each lap?

Validity of the findings

• I have no issues with the validity of the findings. The findings seem sound.

Additional comments

• This is generally a well-written manuscript. I recommend this paper to be accepted with minor revisions.

Reviewer 2 ·

Basic reporting

Summary
The study aimed to investigate the effects of an EMG-driven exoskeleton on the metabolic cost of walking in outdoor conditions. The study included twelve able-bodied participants who wore a powered exoskeleton and completed walking trials on a treadmill and outdoors under various conditions, including unpowered, powered and no exoskeleton. The researchers measured metabolic cost of transport amongst other parameters. The results showed that there was no significant difference in the metabolic cost of transport between powered and unpowered walking conditions with the exoskeleton. The study also found that walking speed was negatively correlated with the cost of transport.

Language
The manuscript was well written and easy to understand. It is technically accurate and conforms to technical standards.

Literature References & Background
The primary objective of the study is to reduce metabolic costs through the use of exoskeletons. The introduction effectively aligns with the research hypothesis, and several contemporary studies that are relevant to the topic are cited.

Article Structure & Raw Data provided
• The article follows a standard structure of introduction, methods, results, discussion, and conclusion, with a coherent storyline and clear explanations provided at appropriate points.
• The figures did not have a good quality in the review document. Please ensure that they are provided in vector graphics format. Additionally, figures 6 and 7 should include standard deviations in a manner consistent with figures 4 and 5.
• The online data is comprehensive but lacks proper documentation such as a readme file. It is unclear why the Metabolic data displays subject numbers as S1-S4 instead of phd13-phd17.

Self-contained with relevant results to hypotheses
The study's hypothesis predicted a decrease in the cost of transport while walking with the powered exoskeleton. However, the results only showed a minor reduction in the cost of transport when comparing powered walking with the exoskeleton to unpowered walking, which did not reach statistical significance. Despite this, the study is self-contained and stands on its own merits.

Experimental design

Research Question is well defined, relevant & meaningful
Previous studies have demonstrated the ability of state-driven exoskeletons to reduce the cost of transport in real-world scenarios, but EMG-driven exoskeletons have not been extensively tested outside of treadmill environments. This study specifically addresses this gap in research. However, the observed reduction in metabolic costs during treadmill walking was considerably lower than in the studies used for comparison.

Rigorous investigation performed to a high technical & ethical standard.
In line 128, it is reported that the subject's age is "22.6 (8.66) years," but it would be helpful to include the minimum and maximum ages as well. If this age range is correct and includes minors, it would be important to explain any specific reasons for including minors in the study, as working with minors often involves additional ethical considerations. From a technical standpoint, the study and data processing are sound. However, I recommend placing more emphasis in the methods section (lines 145-148) on the controller design in the study. Performance downgrades compared to compared to previous studies (such as Jackson & Collins, 2019) should be explained or justified.


Methods described with sufficient detail & information to replicate
The methods section provides detailed step-by-step instructions on how the results were obtained. However, further elaboration is needed on the "subject-specific gain" (line 144). To enhance reproducibility, I recommend creating a code repository that includes the interaction with the powered prosthesis and the calculations used to determine the subject-specific gains. In line 146 it was stated that the exoskeleton is “transparent during swing”: Was the swing phase specifically detected or is this solely based on the Soleus’ EMG signal?

Validity of the findings

Impact and novelty not assessed. Meaningful replication encouraged where rationale & benefit to literature is clearly stated.
This is not a replication study.

All underlying data have been provided; they are robust, statistically sound, & controlled.
As previously mentioned, the data is available, but there is a need for further work to improve its documentation and accessibility. I recommend creating a source code repository that includes both the code for the exoskeleton interaction and the processing of the data. Additionally, it would be helpful to provide documentation for the processed data to enhance its reproducibility and usability for future research.

Conclusions are well stated, linked to original research question & limited to supporting results.
The conclusion section highlights that most of the results were non-significant. It was stated in lines 350-351: „ The value of this study to the literature is that it emphasizes the importance of both hardware dynamics and controller design towards reducing metabolic cost of transport with robotic ankle exoskeletons.” However, the methods section of the paper provides little emphasis on the topic of controller and hardware design. From my perspective, the study demonstrates that using a commercial exoskeleton using a myoelectric controller based on soleus muscle activation solely to enhance the soleus function is insufficient to produce a significant reduction in the cost of transport. Therefore, I recommend further specification of the statement in lines 350-351. Furthermore, the conclusion section does not provide a clear conclusion or interpretation of the results, just a repetition of the hypothesis and the results. Therefore, I recommend the authors add a clear conclusion statement.

Additional comments

Major
I would recommend a minor revision to improve the structure of the available data and provide further clarification on the subject-specificity and streamline the reporting of the results.

There is no mention of the human pacer in the study design. Please include it in the experiment description in the methods section.

It appears that some of the results that were gathered were not reported in the paper. For example, in line 279, there is no mention of what the peak torques were in the outside condition. Similarly, in lines 199-207, there is no information provided regarding the cost of transport for each of the outdoor conditions.

The finding in line 205, indicating a negative correlation between the cost of transport and walking speed, has been previously reported in studies such as "Predicting Metabolic Cost of Level Walking (M. Y. Zarrugh I and C. W. Radcliffe)". Therefore, I suggest that the authors address this in the discussion section.

Minor
Throughout: the use of subjects and participants is not consistent. Please choose one (preferably participants)

Line 127: Why is Female capitalized and male not?

In line 128: “age: 22.6 (8.66) years, height: 1.73 (0.751) m”: The standard deviations seem unlikely. Please doublecheck.

Line 343: No instead of not?

---

## Round 0.2 · Minor Revisions

We would like to see one more quick revision, based on the remaining comments from reviewer 2. Most likely we can then make an editorial decision without additional reviewer input.

Reviewer 2 ·

Basic reporting

Overall, the authors have made commendable efforts to address our concerns. The shared data has been improved with consistent naming and accompanying documentation, which enhances its usability. However, we note that the processing code is still pending upload.

In terms of the subject-specific gains, the authors have provided a more detailed description, which is appreciated. However, there remains a lack of clarity regarding the determination of these gains, particularly in the context of that they were estimated during unpowered walking. In detail (lines 149-151): "Participant-specific static gains were determined during an Unpowered walking condition, and ensured the controlling current reached the maximum value of 7.6 A". How was the static gain determined via the controlling current when the walking condition was unpowered?

Experimental design

The experimental design has not changed in the revised version. The study design was thorough and adequate.

Validity of the findings

I would like to express appreciation for the inclusion of the new Figure 7 in the revised manuscript. This addition provides valuable context, which now makes the observed lower metabolic cost reduction plausible. The inclusion of this figure strengthens the overall argument and improves the clarity of the research findings.

Additional comments

Line 170: The sentence “The exoskeletons only assisted in plantarflexion and were transparent during swing.“ is now unneeded since there is a more detailed explanation above.

The usage of upper case or lower case for the conditions “Powered” and “Unpowered” is inconsistent.

---

## Round 0.3 · accepted · Accept

Thank you for taking care of the final revision. Congratulations!